# Versican G1 Fragment Establishes a Strongly Stabilized Interaction with Hyaluronan-Rich Expanding Matrix during Oocyte Maturation

**DOI:** 10.3390/ijms21072267

**Published:** 2020-03-25

**Authors:** Eva Nagyova, Antonietta Salustri, Lucie Nemcova, Sona Scsukova, Jaroslav Kalous, Antonella Camaioni

**Affiliations:** 1Institute of Animal Physiology and Genetics, Czech Academy of Sciences, 27721 Libechov, Czech Republic; nemcova@iapg.cas.cz (L.N.); kalous@iapg.cas.cz (J.K.); 2Department of Biomedicine and Prevention, University of Rome Tor Vergata, 00133 Rome, Italy; salustri@med.uniroma2.it (A.S.); camaioni@med.uniroma2.it (A.C.); 3Institute of Experimental Endocrinology, Biomedical Research Center, Slovak Academy of Sciences, 84505 Bratislava, Slovakia; scsukova@hotmail.com

**Keywords:** versican, hyaluronan, oocyte-cumulus complex, extracellular matrix

## Abstract

In the mammalian ovary, the hyaluronan (HA)-rich cumulus extracellular matrix (ECM) organized during the gonadotropin-induced process of oocyte maturation is essential for ovulation of the oocyte-cumulus complex (OCC) and fertilization. Versican is an HA-binding proteoglycan that regulates cell function and ECM assembly. Versican cleavage and function remain to be determined in ovarian follicle. We investigated versican expression in porcine ovarian follicles by real-time (RT)-PCR and western blotting. The aims of the present work were to determine whether 1) versican was produced and cleaved by porcine OCCs during gonadotropin stimulation; 2) these processes were autonomous or required the participation of mural granulosa cells (MGCs); and 3) versican cleavage was involved in the formation or degradation of expanded cumulus ECM. We demonstrate two cleavage products of G1 domain of versican (V1) accumulated in the HA-rich cumulus ECM. One of them, a G1-DPEAAE N-terminal fragment (VG1) of ~70 kDa, was generated from V1 during organization of HA in in vivo and in vitro expanded porcine OCCs. Second, the V1-cleaved DPEAAE-positive form of ~65 kDa was the only species detected in MGCs. No versican cleavage products were detected in OCCs cultured without follicular fluid. In summary, porcine OCCs are autonomous in producing and cleaving V1; the cleaved fragment of ~70 kDa VG1 is specific for formation of the expanded cumulus HA-rich ECM.

## 1. Introduction

In mammals, growth and development of the ovarian follicles requires a coordinated series of events controlled by circulating gonadotropins (follicle-stimulating hormone, FSH, and luteinizing hormone, LH) and locally produced growth factors leading to somatic cell proliferation, oocyte development, and steroidogenesis [1]. The cumulus matrix newly organized in oocyte-cumulus complex (OCC) is essential for successful ovulation and fertilization [2,3]. This expanded cumulus extracellular matrix (ECM) is formed by interaction of high molecular weight hyaluronan (HA) and at least three proteins: the serum-derived inter-alpha-trypsin inhibitor (IαI), and two factors produced by cumulus cells, tumor necrosis factor-alpha-induced protein 6 (TNFAIP6) and pentraxin 3 (PTX3) [4,5,6]. Deletion of these corresponding genes in mice produces female infertility due to disorganization of the oocyte-cumulus ECM [3]. Another HA-binding component which may affect the stability of the oocyte-cumulus ECM is the proteoglycan versican [7,8,9]. This proteoglycan has a complex structure constituted by a high-affinity HA binding N-terminal globular domain (G1), a chondroitin sulfate substituted midsection (αGAG and βGAG), and a cell surface- and matrix-interacting C-terminal globular domain (G3) [10]. Alternative splicing events of a single gene [11] generate either a full-length protein possessing both αGAG and βGAG domains (V0~370 kDa), or shorter proteins with the αGAG alone (V2~180 kDa), or the βGAG alone (V1~263 kDa), or neither GAG domains (V3~74 kDa). These forms of splicing produce different effects in the cell environment [12,13,14,15]. In addition, versican is a preferred substrate of specific proteases, a disintegrin and metalloproteinase with thrombospondin like repeats (ADAMTS) [16]. A subset of them, including ADAMTS1, ADAMTS4, and ADAMTS5, cleave versican V0 and V1 in the βGAG domain generating HA-binding fragments ending with the neoepitope DPEAAE at the C-terminus. The degradation product of V0 has large size consisting of a fragment of ~220 kDa carrying several GAG side chains linked to the αGAG domain, while cleavage of V1 generates a small fragment of ~70 kDa consisting mainly of the HA-binding region (G1) of the core protein without GAG side chains attached [16]. The versican G1-DPEAAE, also called versikine [17], is a biologically active molecule in the cell environment affecting apoptosis and angiogenesis during development and tissue remodeling [16,18]. Its fundamental role in the organization of provisional HA matrix is supported by the evidence that mice null for versican as well as for HA synthase 2 presented an identical embryonic lethality due to a defect in the developing of heart matrix [19]. Moreover, analysis of whole rodent ovaries showed upregulated expression levels of versican V0/V1 and their degradation enzymes ADAMTS1 and ADAMTS4 after LH ovulatory surge [8,20]. In agreement, human granulosa cells stimulated in vivo contained abundant *ADAMTS1* and *Versican* mRNA [21], and also porcine OCCs significantly up-regulated the *ADAMTS1* mRNA level [22]. However, versican cleavage and function in ovarian follicle remain to be determined, since it is not known whether cumulus cells are independent in generating versican fragments or require the stimulus or direct participation of granulosa cells. The follicular fluid (FF) contains the heavy chains (HCs) of Iαl that are covalently linked to HA during formation of HA-rich cumulus ECM [23]. It has been proposed that complexes of HC (of IαI)-HA play a central role in matrix organization because they serve as docking sites for the attachment of other proteins to support matrix stabilization [24]. 

The aims of the present work were to determine whether versican was produced and its G1 domain cleaved by porcine cumulus cells, whether these processes were autonomous or required the participation of mural granulosa cells (MGCs), and whether versican cleavage was involved in the formation or degradation of the porcine oocyte-cumulus ECM.

## 2. Results

### 2.1. Versican Fragments Containing the Neo-Epitope DPEAAE in OCCs Expanded in Vivo

In order to detect the DPEAAE-positive species of the N-terminal versican fragments generated by ADAMTS cleavage in the ECM of in vivo expanded porcine OCCs, we performed western blot analyses. The previous published experiments performed with purified molecules showed that the N-terminal fragment of ADAMTS-cleaved V1 does not carry GAGs, while that from V0 has an elevated molecular weight for the presence of several chondroitin sulfate chains and does not enter the gel. Therefore, in order to detect both core protein fragments (~220 and ~70 kDa for V0 and V1, respectively), the OCCs expanded in vivo were isolated from naturally cycling gilts and treated with ChABC. The total extracts were resolved by SDS-PAGE under reducing conditions and probed with the DPEAAE neo-epitope antibody. The results reported in Figure 1 clearly show a positive band of ~65 kDa, approximately the expected molecular weight of G1 fragment from V1 (VG1), while no positivity was detected in the gel at higher molecular weight. However, an evident positive signal was present at the top of the gel, indicating that some versican fragments were prevented from entering. Interestingly, when the HA-rich ECM of the in vivo matured OCCs was digested with Hyal, the material from the top of the gel disappeared and a new positive band appeared in the gel at ~70 kDa. This suggests that a highly stabilized linkage, resistant to denaturating and reducing conditions, occurs between the ~70 kDa DPEAAE-positive versican fragments and HA. This species was mostly present in the matrix extract but also detectable in the cell fraction of in vivo stimulated OCCs, suggesting that it is strongly associated to either the intercellular or pericellular compartment. The intensity of the ~65 kDa band was similar in ChABC and Hyal digested protein extracts indicating that this versican fragment, as the ~70 kDa, does not carry GAGs. In conclusion, V1, but not V0, cleavage products were generated by ADAMTS during OCC expansion. A very faint positive band of ~150 kDa also appeared in the matrix extract after Hyal digestion, but its identity is unknown. 

### 2.2. Versican Fragments Produced by Gonadotropin-Stimulated OCCs Cultured in Vitro

The process of cumulus expansion during oocyte maturation is induced in vitro by gonadotropin stimulation. To verify whether porcine OCCs upregulate versican V1 during in vitro cumulus expansion, as it occurs in vivo, we cultured OCCs in FBS-supplemented medium in the absence or in the presence of FSH/LH. The results reported in Figure 2 showed that the cumulus expansion induced by FSH/LH (Figure 2A) was accompanied by an increase in *VCAN V1* mRNA levels in OCCs analyzed by real-time (RT)-PCR (Figure 2B). A significant increase in the expression was detected at 4 h, peaked at 8-16 h, and returned toward the basal level at 26 h. Then, we investigated whether versican V1 was cleaved in vitro, like in vivo, during the process of cumulus expansion. Groups of porcine OCCs were cultured as above for 44 h (i.e., the time required for in vivo and in vitro full OCC expansion), as shown in Figure 2A. Complexes not stimulated (unexpanded) or stimulated with FSH/LH (expanded) were digested with Hyal or ChABC and analyzed by western blot (Figure 2C). No versican-derived fragments were detected in protein extracts of unstimulated OCCs. In agreement with the results obtained from in vivo expanded OCCs, the complexes stimulated to expand in vitro showed the ~70 kDa versican fragment in the cumulus ECM only after the digestion with Hyal. Moreover, as observed in vivo, the ~65 kDa band of versican fragment was present in these samples and did not require Hyal digestion to enter the gel, since it was present also after ChABC digestion. However, in this in vitro condition, the ~65 and ~70 kDa bands showed similar intensities. We conclude that OCCs induced to expand in vitro with gonadotropins (FSH/LH) in serum-supplemented medium increase the versican V1 expression and accumulate its N-terminal cleaved forms in the HA-rich cumulus ECM, closely resembling the in vivo condition.

### 2.3. Temporal Pattern of Versican Cleavage during in Vitro Expansion 

The follicular fluid (FF) constitutes the physiological environment in which cumulus expansion occurs. Therefore, with the aim to investigate the time course of versican cleavage in the cumulus ECM, groups of 15 porcine OCCs were stimulated for 26 or 44 h with FSH/LH in the presence of FF (Figure 3A). Then, they were digested with Hyal, and matrix and cell extracts were analyzed by western blot. The results clearly indicated that the versican cleaved product of ~70 kDa was already present in the ECM of OCCs cultured for 26 h, and its amount remained apparently unchanged at 44 h. Conversely, the ~65 kDa species accumulated in the cumulus ECM over time, being barely visible at 26 h but quite evident at 44 h. It is known that OCCs stimulated with gonadotropins in the absence of serum or FF did not organize HA-rich cumulus ECM [5]. Importantly, in these culture conditions, no versican fragments were associated with OCCs (Figure 3A, lanes 1 and 2). 

Localization of N-terminal fragments of V1 in the expanded cumulus ECM was analyzed by immunostaining. The anti-DPEAAE antibody showed a bright punctuated positivity throughout the matrix of gonadotropin-stimulated porcine OCCs (Figure 3C), while no signal was detected in the unstimulated OCCs (Figure 3B). 

### 2.4. Versican Fragments Produced by Mural Granulosa Cells after in Vitro Gonadotropin Stimulation 

First, we analyzed by RT-PCR whether porcine MGCs, as cumulus cells, upregulated *VCAN V1* mRNA following in vitro gonadotropin stimulation. Mural GCs were isolated from naturally cycling gilts and incubated in culture medium supplemented with 5% FBS in the presence or absence of FSH/LH. Results reported in Figure 4A showed that gonadotropin-stimulated MGCs, unlike those unstimulated, transiently increased *VCAN V1* expression. The increase was statistically significant starting from 8 h of culture and followed a temporal pattern similar to that observed in OCCs.

Then, we analyzed the pattern of V1 cleavage by MGCs during in vitro culture. To this purpose, MGCs freshly isolated (0 h) or cultured for 44 h in medium supplemented with FBS and FSH/LH were digested with either ChABC or Hyal, and the extracts were analyzed by western blot using the anti-DPEAAE neo-epitope antibody (Figure 4B). Interestingly, both enzymatic digestions gave the same pattern, with only the ~65 kDa versican species present in the total and matrix extracts (Figure 4B; lanes 1–3, 5, and 8). The ~70 kDa V1 fragment found in cumulus ECM was not detected in the cultured MGCs, even after Hyal digestion.

## 3. Discussion

The ECM organized by cumulus cells around the mammalian oocyte during oocyte maturation is essential for oocyte release from the follicle, its transfer to the oviduct, and for sperm migration and fertilization [3,25,26]. In the present study, we demonstrate for the first time that the cumulus ECM contains two types of DPEAAE-positive G1 fragments derived from versican 1 (VG1): one is specific to cumulus cells while the other is in common with granulosa cells. 

Previously, it has been shown that the ADAMTS family of proteinases cleave proteoglycan versican, and its fragments have distinct functions in matrix organization [27]. In order to detect the DPEAAE-positive species of the VG1 fragments generated by ADAMTS cleavage in in vivo expanded porcine OCCs, we performed western blot analyses. We found that a product of cleavage, DPEAAE-positive ~70 kDa fragment of VG1, entered into the gel only when OCCs were digested with Hyal. This suggests that a highly stabilized linkage, resistant to denaturating and reducing conditions, occurs between the ~70 kDa DPEAAE-positive versican fragments and HA. This observation provided evidence that disruption of disulfide bonds per se was not sufficient, and enzymatic HA degradation was necessary to release the fragment from the cumulus ECM. Our results suggest that the link module in the G1 domain is not the only domain involved in the integration of the V1 fragment into the cumulus ECM, but that there are present supplementary interactions with HA and other molecules.

To clarify whether the in vitro process mimics the in vivo situation, both in terms of gene expression and protein cleavage, porcine OCCs were cultured with gonadotropins in the presence of serum or follicular fluid. Both the latter are necessary as a source of IαI, which provides the HCs that are covalently linked to the synthesized HA during organization of cumulus ECM. Under these in vitro culture conditions, porcine OCCs up-regulated *VCAN V1* mRNA and generated a ~70 kDa VG1 fragment. Further, we followed in vitro the cleavage of versican by OCCs over time and found that the ~70 kDa VG1 fragment was clearly evident at 26 and 44 h of culture. The time course of V1 cleavage correlates with the expression and activation of *ADAMTS1*, reported by Shimada et al. [22]. Our observations that the ~70 kDa VG1 fragment associates to HA during formation of the cumulus ECM strongly support the hypothesis that synthesis of V1 and its processing is related to proper HA matrix organization. A role of the VG1 fragment in HA-rich cumulus ECM organization is reinforced by findings that a versican fragment is able to bind to HCs of IαI [18]. The HCs are transferred from the chondroitin sulphate of IαI to HA through a trans-esterification reaction and become covalently linked to HA during cumulus expansion in all analyzed species [5,23,28]. The temporal pattern of HCs (of IαI) incorporation into the cumulus ECM [6] was similar to the ~70 kDa VG1 fragment here. In agreement with the possible interaction of these two types of components in the cumulus ECM, we found no versican cleavage products in the porcine OCCs stimulated in vitro in the absence of FF, i.e., without a source of HCs of IαI. The immunofluorescence analysis on expanded OCCs with the anti-G1 DPEAAE antibody showed a punctuated positivity in the cumulus, suggesting the presence of VG1 fragments. In agreement, Murasawa et al. [18] identified a macromolecular complex formed by VG1 aggregates associated to HCs (of IαI) translocated onto HA as provisional matrices formed in inflamed skin. As second cleavage product, we found a lower molecular weight VG1 fragment (~65 kDa) in the OCC matrix extract that entered into the gel independently from Hyal treatment. It was present in OCCs expanded in vivo and at 44 h in gonadotropin-stimulated OCCs, when the cumulus expansion was completed and remodeling of cumulus ECM began. This lower molecular weight fragment is likely the result of further degradative processes taking place at the time of ovulation, when massive synthesis and activation of several hydrolytic enzymes occur in granulosa and cumulus cells [29], leading to the detachment of OCC from the wall and follicle rupture. Since the fragment is recognized by the antibody against the DPEAAE at the C-terminus, it likely derives from the proteolysis of VG1 N-terminus, a G1 subdomain which is known to enhance binding capacity of versican to HA [30]. Then, the N-terminus degradation of VG1 fragment can determine its decreased ability to become firmly associated to the HA in the cumulus ECM. Finally, MGCs cultured in vitro under the same conditions for 44 h produced only the ~65 kDa VG1 fragment. Accordingly, MGCs do not produce HA and release versican into the FF in a form unable to bind to HA [31,32].

In summary, we provide evidence that porcine OCCs are autonomous in producing and cleaving Versican 1 (V1) during the process of oocyte maturation. We show two distinct cleavage products of G1 domain of V1 accumulated in the cumulus ECM. One of them, the cleaved fragment of ~70 kDa VG1, interacted strongly with the HA-rich cumulus ECM. As second cleavage product, VG1 fragment (~65 kDa) was the only one species detected in MGCs. 

Results reported in the present study strongly suggest that this VG1 fragment is involved in stabilizing the HA-rich cumulus ECM structure by interacting with HA-HCs (of IαI) complexes. Since infertility is a serious health problem in females, to clarify molecular and cellular mechanisms in ovarian follicles is an urgent need. The results reported here could help in diagnosis and therapy of female infertility.

## 4. Materials and Methods 

### 4.1. Isolation and Culture of Oocyte-Cumulus Complexes and Mural Granulosa Cells

Porcine ovaries were collected at a local abattoir and immediately transported to the laboratory. Oocyte-cumulus complexes (OCCs) were aspirated from medium-sized antral follicles about 3–5 mm in diameter. The culture medium was M199 (Gibco, Invitrogen) supplemented with 20 mM NaHCO_3_, 6.25 mM HEPES, 0.91 mM sodium pyruvate, 1.62 mM calcium lactate, and antibiotics (all from Sigma, Prague, Czech Republic) as well as with recombinant human follicle-stimulating hormone (FSH; 100 ng/mL; Gonal, Merck Serono, Modugno, Italy), recombinant human luteinizing hormone (LH; 100 ng/mL; Luveris, Merck Serono, Modugno, Italy), and 5% fetal bovine serum (FBS; Sigma-Aldrich, Schnelldorf, Germany) or PVP (polyvinylpyrolidone; 3 mg/mL; Sigma, Prague, Czech Republic). Groups of 30 OCCs were cultured in 300 µL of medium in 4-well dishes (Nunclon, Roskilde, Denmark) at 38.5 °C in an atmosphere of 5% CO_2_ in air for 26 or 44 h.

Mural granulosa cells (MGCs) were aspirated from medium-sized porcine ovarian follicles (3–5 mm in diameter). Fifteen clumps of MGCs were incubated in the same culture conditions as above for 44 h.

### 4.2. RNA Isolation

Total RNA from 30 OCCs or 15 clumps of MGCs either unstimulated (controls) or stimulated in vitro with FSH/LH (2, 4, 8, 16, 26, and 44 h) were extracted using an RNeasy Mini Kit (Qiagen, Hilden, Germany) following the manufacturer’s instructions. RNA was eluted in 50 μL of RNase free H_2_O. After isolation, the concentrations and quality of RNA in samples were assessed by a Nanodrop ND-1000 spectrophotometer (NanoDrop Technologies, Wilmington, DE, USA). Concentrations of RNA in all samples were adjusted to 15 ng/μL. All RNA samples were frozen and stored at −80 °C.

### 4.3. Real-Time (RT)-PCR

The quantitative reverse transcription polymerase chain reaction was carried out using a One-Step RT-PCR kit (Qiagen) and specific oligonucleotide primers for pig *VCAN V1* (forward: 5′-GCCTA CTG CTA TAA ACG TCG AAT-3′, reverse: 5′-TAG TCG TGA CGT CAG TGG CA-3′, AB558521, product length: 217 bp) and (forward: 5′-CCA GTA AAC GGG CGA TAT AA-3′, reverse: 5′-CTT GAC CAA GGA AAG CAA GG-3′, NM_001032376, product length: 129 bp). To reduce possible DNA contamination, primers spanning exon–exon junctions (*VCAN V1*) or two introns (*HPRT1*) were designed using Beacon Designer software (Premier Biosoft, CA, USA). The total RNA of the samples was reverse-transcribed and subsequently amplified in a 25 µL reaction mixture containing 5 µL of 5× reaction buffer, dNTP (each 400 µM), SybrGreenI (0.5 µL of 1:1000 stock solution, Molecular Probes, Eugene, Oregon), primers (each 400 µM), RNasine inhibitor (5 IU; Promega, Madison, WI, USA), Qiagen One-Step RT-PCR enzyme mix (1 µL) and total RNA (2 µL). The amplification was performed on a RotorGene RG-3000 cycler (Cobett Research, Sydney, Australia). The reaction conditions were as follows: cDNA synthesis at 50 °C for 30 min, initial activation at 95 °C for 15 min, and cycles of denaturation (95 °C for 15 s), annealing (55 °C for 15 s), extension (72 °C for 20 s), and final extension (72 °C for 5 min). Fluorescence data were acquired during an addition step at approximately 3 °C below the melting temperature of the product. After the cycling, the melting curves were generated to verify the amplification of one specific target in each tube. In addition, the specificity of RT-PCR products was assessed by gel electrophoresis on 1.5% agarose gel with MidoriGreen Direct (Nippon Genetics, Dueren, Germany) staining. The relative concentration of templates in different samples was determined using comparative analysis software (Corbett Research). The ratio of the target gene concentration to reference gene (*HPRT1*) mRNA concentration was estimated in each sample.

### 4.4. Western Blot Analysis 

Porcine oocyte-cumulus complexes (OCCs) expanded in vivo or compact OCCs or mural granulosa cells (MGCs) stimulated to expand in vitro with FSH/LH were digested with chondroitinase ABC (ChABC) or *Streptomyces* hyaluronlyticus (Hyal; both; Merck- Calbiochem, Prague, Czech Republic) and analyzed by western blot as previously described [5,6]. Shortly, complexes obtained after in vivo or in vitro stimulation were digested with 1U of Hyal or ChABC (50 mU) in 20–30 µL of PBS-PI at 37 °C for 2 h. In some experiments, after digestion, the samples were centrifuged at 300× g for 5 min to divide the total extract (T) into the two compartments: the matrix (M) and the cell (C). All samples were then extracted by adding reducing Laemmli buffer, boiled at 100 °C for 4 min, separated in 7.5% acrylamide/SDS gels, and transferred to Hypobond –P membranes. Total, matrix and cell extracts were probed using a versican antibody (ab19345 Abcam, Prague, Czech Republic) that recognized the neo-epitope DPEAAE generated at the N-terminal of V1 following ADAMTS1 digestion; these N-terminal fragments had an apparent MW of 65–70 kDa. Actin (A2066; Sigma, Prague, Czech Republic) immunodetection was performed to compare sample loading. 

### 4.5. Immunofluorescence Analysis

The non-expanded and expanded OCCs were collected after 26 h of culture in FBS (control) or FBS plus FSH/LH-supplemented medium, respectively. The expanded complexes were fixed in 4% paraformaldehyde for 30 min at room temperature, permeabilized in 0.1% Triton/PBS for 30 min, and blocked in 5% normal goat serum/PBS for 1 h at room temperature. Subsequently, OCCs were incubated with primary anti-versican antibody (ab19345, Abcam, Prague, Czech Republic; 1:100,) for 72 h at 4 °C followed by incubation with Alexa Fluor 488 goat anti-rabbit IgG as the secondary antibody (Molecular Probes, Europe BV, Leiden, the Netherlands; 1:500) for 1 h at room temperature. DNA was detected with 2.5 mg/mL of 4,6-diamidino-2-phenylindole (DAPI). Finally, the OCCs were mounted on glass slides in mounting medium and examined with an Olympus AX70 microscope equipped with a DP30BW CCD camera. Image files were edited with Adobe Photoshop computer software.

### 4.6. Statistical Analysis

The *VCAN V1* expression data were obtained from three independent experiments. Gene expression between stimulated and unstimulated OCCs at each time point of culture was compared by the Student’s t-test. The differences in relative concentration of templates during culture times were analyzed by ANOVA followed by Tukey’s post-test using SigmaStat 3.0 software (Jandel Scientific, San Rafael, CA, USA).

## Figures and Tables

**Figure 1 ijms-21-02267-f001:**
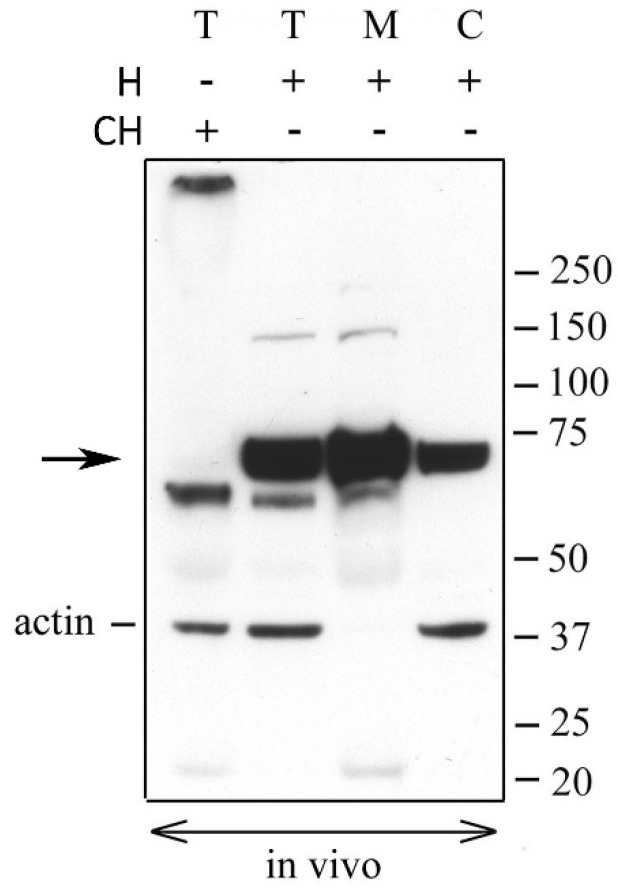
Versican fragments containing the neo-epitope DPEAAE are detected in OCCs expanded in vivo. The complexes expanded in vivo were isolated from naturally cycling gilts (two porcine OCCs for each lane). Total (T), matrix (M) and cell (C) extracts from expanded OCCs digested with *Streptomyces* hyaluronidase (H) or chondroitinase ABC (CH) were analyzed by western blot with anti-DPEAAE versican antibody. Actin immunodetection was performed to compare sample loading. A representative blot of four independent experiments is shown.

**Figure 2 ijms-21-02267-f002:**
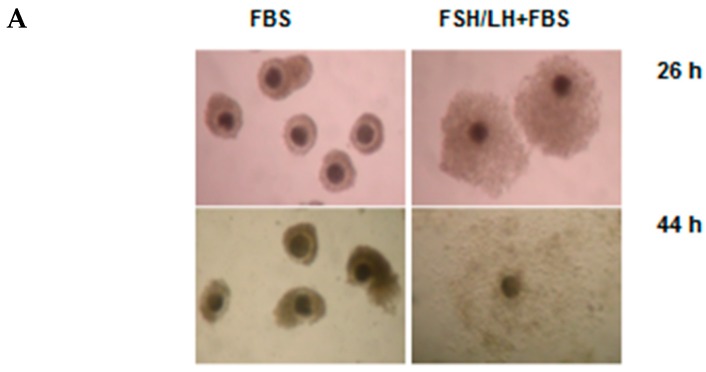
Effects of gonadotropins on cumulus expansion and versican expression oocyte-cumulus complexes (OCCs) cultured in vitro. (**A**) Morphology of OCCs stimulated with gonadotropins (100 ng/mL follicle-stimulating hormone (FSH) + 100 ng/mL luteinizing hormone (LH)) cultured in vitro for 26 and 44 h in medium supplemented with 5% FBS. Note that the size of OCCs cultured in the presence of gonadotropins was larger than that of OCCs cultured only in basal medium (FBS alone). Full expansion was reached at 44 h when cumulus cells became scattered in an abundant extracellular matrix (ECM). (**B**) *Versican V1* mRNA (*VCAN V1*) expression investigated by RT-PCR in hormone-stimulated OCCs cultured in vitro. Groups of 30 OCCs were immediately processed (0 h; white bar) or cultured in FBS-supplemented medium without (light grey bars) or with (grey bars) FSH/LH stimulation, as above, for the indicated time points. Three independent experiments were performed. Bars show means ± SEM. In the time course of gonadotropin-stimulated OCCs; different superscripts (a–c) indicate statistically significant differences compared to 0 h sample (*p* < 0.05). Asterisks indicate statistically significant differences between unstimulated and stimulated groups of OCCs at the same time point (** *p* < 0.01, *** *p* < 0.001). (**C**) Versican 1 fragments containing the neo-epitope DPEAAE accumulated in the cumulus matrix of hormone-stimulated OCCs cultured in vitro. Groups of 15 OCCs were cultured for 44 h in FBS-supplemented medium in the absence/presence of FSH/LH. Complexes were digested with *Streptomyces* hyaluronidase (H) or chondroitinase ABC (CH) and total (T), matrix (M), and cell (C) extract from not expanded (without hormones; lanes 1 and 2) or fully expanded (with hormones, lanes 3-8) were analyzed by western blot with anti-DPEAAE versican antibody. Actin immunodetection was performed to compare sample loading. A representative blot of three independent experiments is shown.

**Figure 3 ijms-21-02267-f003:**
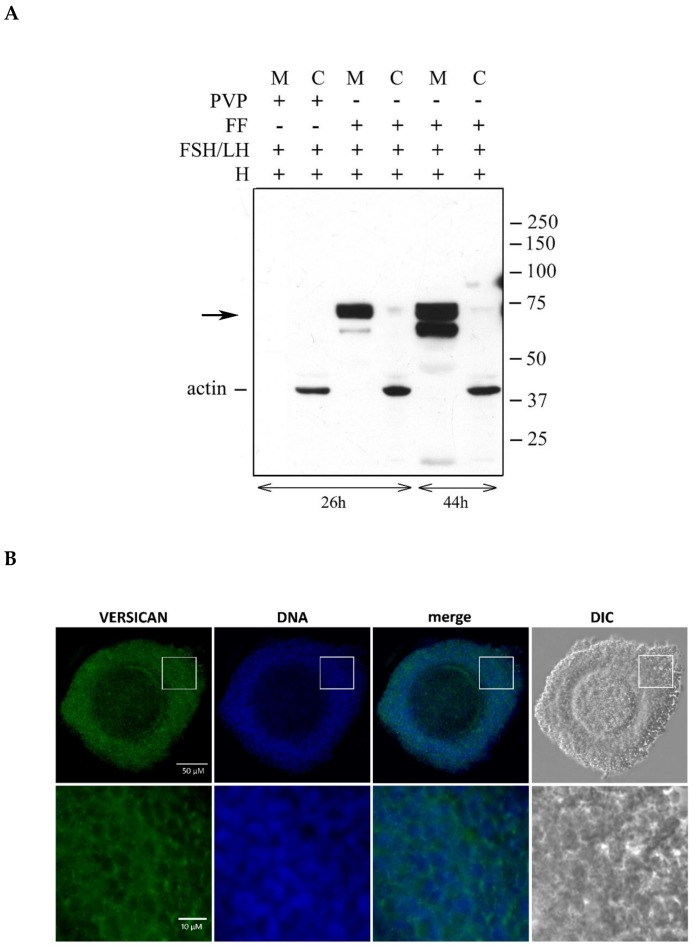
(**A**) Temporal pattern of versican V1 cleavage in hormone-stimulated OCCs cultured in vitro in the absence and in the presence of follicular fluid (FF). Groups of 15 OCCs were stimulated for 26 or 44 h with FSH/LH in the absence (PVP) or presence of 5% FF. At the indicated times of culture, they were digested with *Streptomyces* hyaluronidase (H), and matrix (M) and cell (C) extracts were analyzed by western blot with anti-DPEAAE-V1 antibody. Actin immunodetection was performed to compare sample loading. A representative blot of three independent experiments is shown. Immunostaining of porcine OCCs cultured for 26 h without (**B**) and with (**C**) FSH/LH in the presence of FBS. The OCCs were labeled with anti-DPEEAE versican primary antibody, and with secondary antibody Alexa Fluor 488. Nuclei were co-stained with DAPI. The OCCs were mounted on glass slides and examined with an Olympus AX70 microscope equipped with a DP30BW CCD camera. Note the punctuate positivity only in the matrix of expanded OCCs (panel **C**, higher magnification).

**Figure 4 ijms-21-02267-f004:**
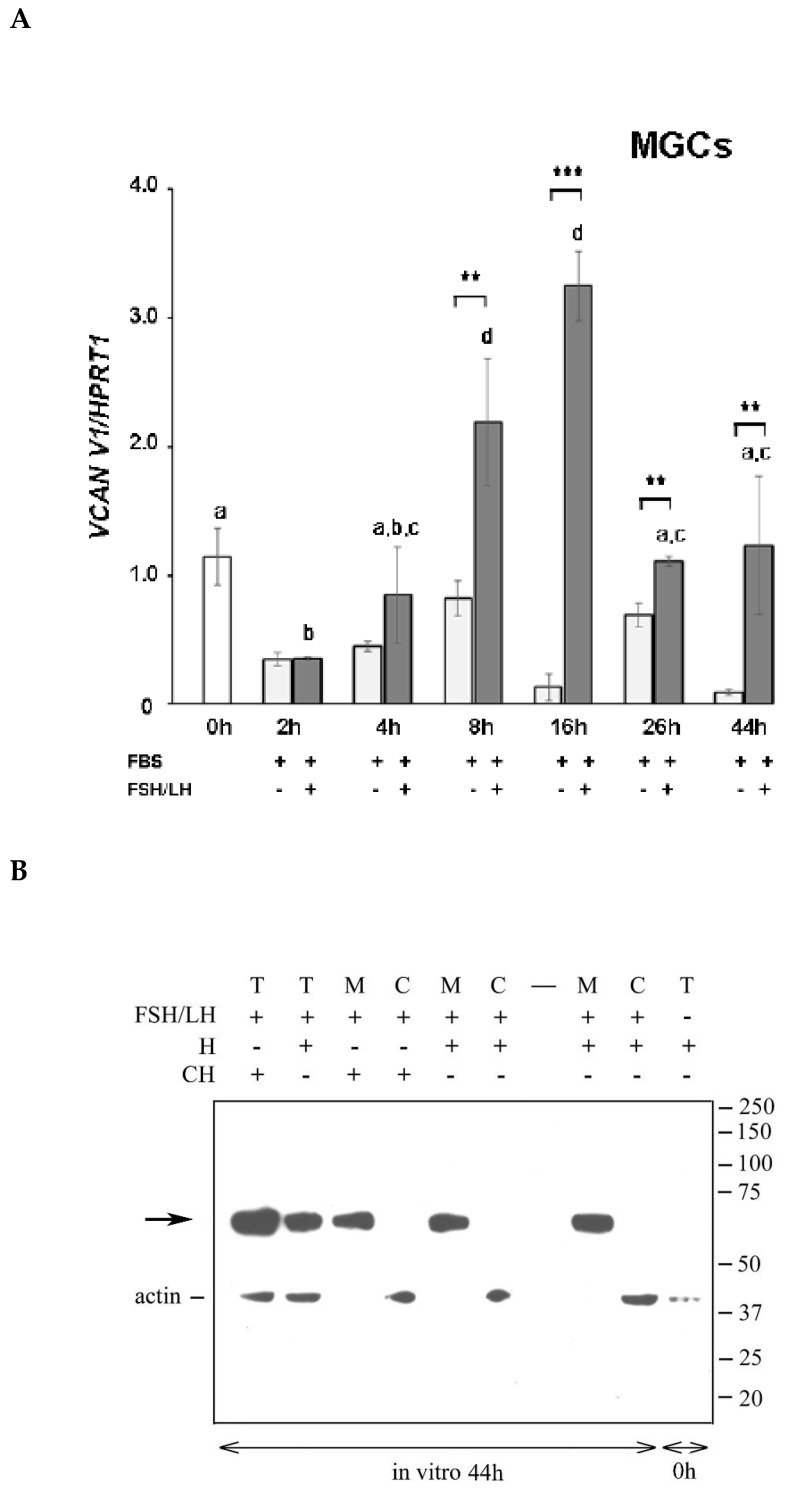
(**A**) *Versican 1* mRNA (*VCAN V1*) expression in hormone-stimulated mural granulosa cells (MGCs) cultured in vitro. Fifteen small pieces of follicle wall consisting of MGC aggregates were either immediately processed (0 h, white bar) or cultured in 5% FBS-supplemented medium without (light grey bars) or with (grey bars) FSH/LH for the indicated time points. The data shown are representative of three independent experiments. Bars show means ± SEM. Different superscripts indicate statistically significant differences compared to 0 h sample (*p* < 0.05). Asterisks indicate statistically significant differences between unstimulated and hormone-stimulated MGCs at the same time point (** *p* < 0.01, *** *p* < 0.001). (**B**) Versican fragments containing the neo-epitope DPEAAE in mural granulosa cells (MGCs) after in vitro stimulation with gonadotropins. Porcine MGCs were collected before (0 h) or after 44 h of culture in 5% FBS-supplemented medium in the presence of FSH/LH. Total (T), matrix (M) and cell (C) extracts from collected MGCs were digested with *Streptomyces* hyaluronidase (H) or chondroitinase ABC (CH) and analyzed by western blot with anti-DPEAAE versican antibody. Actin immunodetection was performed to compare sample loading. A representative blot of three independent experiments is shown.

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
