# Peer review of "Versican G1 Fragment Establishes a Strongly Stabilized Interaction with Hyaluronan-Rich Expanding Matrix during Oocyte Maturation"

_ijms, 2020, doi:10.3390/ijms21072267_

Round 1

Reviewer 1 Report

The authors conducted this study to determine the function of Versican in the regulation of ovarian follicles. It is innovative and it could give answer to various ovarian disorders but not in its current form. My concerns:

  1. The authors are kindly requested to conform to the ARRIVE Statement for animal pre-clinical studies. I cannot tell where the specimens derived from.
  2. The authors are kindly requested to shorten the Introduction section.
  3. The authors are kindly requested to summarize their findings in the first paragraph of the Discussion section, and then to comment on them throughout this section. There is no need for repeat of the already known studies.
  4. The authors are kindly requested to add a clear and concise limitation paragraph, along with a paragraph of the implications of the findings of their study. 

Author Response

Review report 1

Q1.      The authors are kindly requested to conform to the ARRIVE Statement for animal pre-clinical studies. I cannot tell where the specimens derived from.

In our study, all the experiments were performed on porcine samples obtained from a local abattoir. See Materials and Methods, page 15, lines 332-346: “Porcine ovaries were collected at a local abattoir and immediately transported to the laboratory.” These experiments do not require any special permits to work with laboratory animals.

Moreover, in the abstract we have stated the origin of the samples.

See in Abstract: We investigated versican expression in porcine ovarian follicle by real-time RT-PCR and Western blotting. The aims of the present work were to determine whether: 1) versican was produced and cleaved by porcine OCCs during gonadotropin stimulation; 2) these processes were autonomous or required the participation of mural granulosa cells (MGCs); and 3) versican cleavage was involved in the formation or degradation of expanded cumulus ECM.

In summary, porcine OCCs are autonomous in producing and cleaving V1; the cleaved fragment of ~70 kDa VG1 is specific for formation of the expanded cumulus HA-rich ECM.

Q2.      The authors are kindly requested to shorten the Introduction section.

The introduction section has been shortened according to reviewer suggestion. See page 2, lines 34-80.

Q3.      The authors are kindly requested to summarize their findings in the first paragraph of the Discussion section, and then to comment on them throughout this section. There is no need for repeat of the already known studies.

The Discussion section has been adapted according the reviewer suggestion, including the first paragraph summarizing the findings of our study. See page 13, lines 265-270 and also page 15, lines 319-327.

The ECM organized by cumulus cells around the mammalian oocyte during oocyte maturation is essential for oocyte release from the follicle, its transfer to the oviduct and for sperm migration and fertilization [3,25,26]. In the present study, we demonstrate for the first time that the cumulus ECM contains two types of DPEAAE-positive G1 fragments derived from versican 1 (VG1). One is specific to cumulus cells, while the other is in common with granulosa cells.

Q4.      The authors are kindly requested to add a clear and concise limitation paragraph, along with a paragraph of the implications of the findings of their study.

A paragraph describing the implications of the findings of our study has been introduced; see page 15, lines 319-329.

In summary, we provide evidence that porcine OCCs are autonomous in producing and cleaving Versican 1 (V1) during the process of oocyte maturation. We show two distinct cleavage products of G1 domain of V1 accumulated in the cumulus ECM. One of them, the cleaved fragment of ~70 kDa VG1, interacted strongly with the HA rich-cumulus ECM. As second cleavage product, VG1 fragment (~65 kDa) was the only one species detected in MGCs.

Results reported in the present study strongly suggest that this VG1 fragment is involved in stabilizing the HA-rich cumulus ECM structure by interacting with HA-HCs (of IαI) complexes. Since infertility is one of the serious health problems in female, to clarify molecular and cellular mechanisms in ovarian follicles is an urgent need. The results reported here could help in diagnosis and therapy of female infertility.

We thank for your help to clarify better our results concerning versican expression in porcine ovarian follicle. We greatly  appreciate  your perfect suggestions. 

Reviewer 2 Report

The authors stating that 'qualitative changes of WB is enough to interpret the results than the normalized quantitative WB'.

Otherthan that the reviewer do not have any comments.

Author Response

Review report 2

Comments and Suggestions for Authors

The authors stating that 'qualitative changes of WB is enough to interpret the results than the normalized quantitative WB'.

Other than that the reviewer do not have any comments.

We thank for your help to clarify better our results concerning versican expression in porcine ovarian follicle. We greatly appreciate your very useful suggestions. 

Round 2

Reviewer 1 Report

The authors have addressed adequately the concerns of this reviewers.